# Lactate in the Tumor Microenvironment: An Essential Molecule in Cancer Progression and Treatment

**DOI:** 10.3390/cancers12113244

**Published:** 2020-11-03

**Authors:** Ricardo Pérez-Tomás, Isabel Pérez-Guillén

**Affiliations:** 1Cancer Cell Biology Research Group, Department of Pathology and Experimental Therapeutics, Faculty of Medicine and Health Sciences, University of Barcelona, C./ Feixa Llarga s/n, L’ Hospitalet de Llobregat, 08905 Barcelona, Spain; 2Microbiology Unit, Department of Pathology and Experimental Therapeutics, Faculty of Medicine and Health Sciences, University of Barcelona, C./ Feixa Llarga s/n, L’ Hospitalet de Llobregat, 08905 Barcelona, Spain; isaperezguillen@ub.edu

**Keywords:** lactate, tumor microenvironment (TME), acidosis, LDH, MCTs

## Abstract

**Simple Summary:**

The role of lactate in cancer described by Otto Warburg in 1927 states that cancer cells uptake high amount of glucose with a marked increase in lactate production, this is known as the “Warburg effect”. Since then lactate turn out to be a major signaling molecule in cancer progression. Its release from tumor cells is accompanied by acidification ranging from 6.3 to 6.9 in the tumor microenvironment (TME) which favors processes such as tumor promotion, angiogenesis, metastasis, tumor resistance and more importantly, immunosuppression which has been associated with a poor outcome. The goal of this review is to examine and discuss in deep detail the recent studies that address the role of lactate in all these cancerous processes. Lastly, we explore the efforts to target the lactate production and its transport as a promising approach for cancer therapeutics.

**Abstract:**

Cancer is a complex disease that includes the reprogramming of metabolic pathways by malignant proliferating cells, including those affecting the tumor microenvironment (TME). The “TME concept” was introduced in recognition of the roles played by factors other than tumor cells in cancer progression. In response to the hypoxic or semi-hypoxic characteristic of the TME, cancer cells generate a large amount of lactate via the metabolism of glucose and glutamine. Export of this newly generated lactate by the tumor cells together with H+ prevents intracellular acidification but acidifies the TME. In recent years, the importance of lactate and acidosis in carcinogenesis has gained increasing attention, including the role of lactate as a tumor-promoting metabolite. Here we review the existing literature on lactate metabolism in tumor cells and the ability of extracellular lactate to direct the metabolic reprogramming of those cells. Studies demonstrating the roles of lactate in biological processes that drive or sustain carcinogenesis (tumor promotion, angiogenesis, metastasis and tumor resistance) and lactate’s role as an immunosuppressor that contributes to tumor evasion are also considered. Finally, we consider recent therapeutic efforts using available drugs directed at and interfering with lactate production and transport in cancer treatment.

## 1. Introduction

In 1922, Archibald V. Hill and Otto Meyerhoff received a Nobel Prize for their work on the energetics of carbohydrate catabolism in skeletal muscle [1]. Hill had quantified the energy released from the conversion of glucose to lactic acid and determined that glucose oxidation occurs during limited oxygen availability. Meyerhoff elucidated most of the glycolytic pathway and demonstrated that lactic acid is a byproduct of glycolysis, initiated in the absence of oxygen [2]. (For readers interested in the history of lactate, the 2005 review by Phil et al. [3] is recommended). 

Cancer cells have an enormous capacity to regulate their metabolism to promote tumor formation, establishment and survival. In fact, this ability is considered a hallmark of cancer [4]. The increased metabolic rates of several types of neoplasm were first discovered by Otto Warburg in 1927. Warburg showed that neoplastic cells use large amounts of glucose as a primary energy source, even in the presence of oxygen, and thus produce large amounts of lactate, a process referred to as aerobic glycolysis [5]. While less efficient than the full cycle of glycolysis linked to the Krebs cycle and oxidative phosphorylation (oxphos) in terms of ATP production (two ATP per glucose molecule vs. 38 ATP per glucose molecule), aerobic glycolysis generates ATP much faster [6]. In resting normal cells, the glycolytic rate is low and most glucose is converted into carbon dioxide and water. However, in proliferating normal cells, and especially in cancer cells, aerobic glycolysis results in the conversion of as much as 85% of the incoming glucose to lactate [7] (Figure 1). Thus, in 1972, Efraim Racker referred to the high rate of aerobic glycolysis by cancer cells as the “Warburg effect.” It describes the decrease in the oxphos of glucose that supports tumor survival and metastasis [8].

Along with the Warburg effect, glutaminolysis is the most important source of lactate and protons in cancer cells and the cancer environment [8,9]. Glutamine serves as a cell nutrient, entering the cells via the glutamine transporter ASCT2 and the neutral amino acid transporter SLC1A5 [10]. In the mitochondria, glutamine is converted to glutamate, which in turn is converted to α-ketoglutarate and either fed into the Krebs cycle (Figure 1) or used for the production of glutathione, a major thiol-containing endogenous antioxidant and an essential player in tumor cell survival [11].

Although long considered to be a “metabolic waste product,” it is now clear that lactate plays a critical role in fueling the proliferation of tumor cells and in carcinogenic signaling [12], metastasis [13] and tumor resistance [14]. In this review, we discuss the roles of cellular lactate synthesis, transport and exchange in driving and sustaining carcinogenesis, specifically tumor promotion, immune escape angiogenesis, metastasis and tumor drug resistance. These same features suggest that the targeting of lactate production and transport offers a novel approach to the treatment of cancer.

## 2. The Tumor Microenvironment and the Reversed pH Gradient

Large amounts of lactic acid and H^+^ produced during aerobic glycolysis and glutaminolysis are released into the extracellular space, thereby entering the tumor microenvironment (TME) [15]. This complex, dynamic cellular compartment is an integral, essential feature of cancer—a major contributor to the aggressiveness of the disease, a determinant of the therapeutic response, but also itself a potential target for therapeutic intervention [16,17]. It is a heterogeneous niche harboring the physical and physiological components that empower tumor cells to progress and metastasize [4,18]. Among the physical components of the TME are tumor cells, endothelial cells, cancer-associated fibroblasts (CAFs), immune cells, blood vessels, extracellular matrix (ECM), growth factors and cellular metabolites [15]. The physiological components consist of oxygen, pH, nutrients, metabolic products, signaling molecules, reducing/oxidizing species, growth factors and protumorigenic factors. 

Lactate is one of the most significant metabolites in the TME. Whereas physiological concentrations of lactate in blood and healthy tissue are in the range of 1.5–3 mM [19], the release of lactate from tumor cells can result in extracellular concentrations as high as 40 mM [20]. This was demonstrated by Fisher et al. in 2017 [21], who found high levels of lactate in the sera of patients with different types of cancers (breast, gastrointestinal, lung and urogenital cancers; melanoma; sarcoma; etc.). Another important physiological feature of the TME is the hypoxia [22] arising from the imbalanced angiogenesis characteristic of most tumors. Hypoxia induces changes in cellular metabolic pathways, including an even greater dependence on aerobic glycolysis, which together with glutaminolysis increases lactate production.

Furthermore, while under normal physiological conditions, the pH of blood and tissues is tightly controlled at around pH 7.4; in the TME the local pH typically ranges from 5.6 to 7.0 [23]. In fact, nearly all tumors show an increase in intracellular pH (pHi = 7.3–7.7) and a decrease in extracellular pH (pHe = 6.3–6.9) compared to healthy tissue (pHi = 7.0, pHe = 7.4). This “reversed pH gradient” [24,25,26] is a property of all actively growing cells [27], which suggests its evolutionary importance [24]. Numerous processes essential to normal cellular metabolism are highly pH sensitive in the physiologically relevant range. These include the activities of lactate dehydrogenases (LDHs, discussed in detail below) and the gluconeogenic enzyme phosphofructokinase (PFK) [28,29], both of which require an alkaline pH. An increased pHi also enhances cancer cell properties such as proliferation [30] and promotes cell survival by limiting apoptosis [31]. Specifically, the glycolytic flux, essential for metabolic reprogramming, is stimulated by an alkaline cytosol, whereas the caspase-induced activation of apoptosis depends on a mild acidification of the cytosol [32]. 

At the same time, a decreased pHe creates the perfect environment for many of the acquired characteristics of cancers cells, in addition to facilitating tumor immune escape and effective proteolytic degradation of the ECM by invading tumor cells [33,34]. ATP hydrolysis coupled with glycolysis is the main source of the H+ that results in the decreased extracellular pH [35]. The inefficient removal of H+ and lactic acid from the extracellular space, due to the poorly perfused, dense tumor tissue, acidifies the TME and contributes to the reversed pH gradient [36]. The development of this tumor phenotype is an early event in tumorigenesis that becomes more prominent with the progression of cancer, as its maintenance reinforces the metabolic adaptation, survival, invasion, immune evasion and drug resistance of tumor cells. In fact, together with genome instability, a reversed pH gradient can be considered as an underlying cellular requirement in tumorigenesis. The TME is now widely recognized as an integral and essential component of cancer, a major contributor to the aggressiveness and treatment resistance of the disease, but also a potential target for therapeutic intervention [16,17]. 

## 3. Lactate Dehydrogenases

Lactate dehydrogenases (LDHs) are a group of metabolic enzymes that simultaneously catalyze the reversible conversion of pyruvate to lactate and play an important role in cancer metabolism [37]. Of the five LDH isoenzymes described (LDH-1–5), LDH-5, also known as LDHA, and LDH-1, also referred to as LDHB, are of interest in this review.

LDHA preferentially reduces pyruvate to lactate during glycolysis, accompanied by the regeneration of NADH to NAD^+^, hence minimizing pyruvate entry into the Krebs cycle in the mitochondria. Thus, LDHA ensures the maintenance of a “fuel” supply in cancer cells in addition to supporting tumor growth and invasion even under hypoxic conditions. Moreover, LDHA activity can be further enhanced by post-translational modifications, such as the phosphorylation of tyrosine-10, which has been shown to contribute to tumor metastasis by altering cell metabolism [38,39]. The expression of LDHA is regulated by the oncogene c-Myc and by hypoxia-inducing factor 1 (HIF-1) and microRNA miR-34a [40,41]. Associations of the increased expression of LDHA with a lower event-free survival rate and resistance to chemotherapy have been demonstrated in numerous tumors [42]. Furthermore, high-level LDHA expression serves as a prognostic indicator in patients with different cancers [43]. 

LDHB supports the conversion of lactate to pyruvate in cells that use lactate as a nutrient for oxidative metabolism or gluconeogenesis [44]. The protons generated by the enzyme promote V-ATPase-dependent lysosomal acidification and autophagy, creating an additional source of energetic and biosynthetic precursors in metabolically restricted microenvironments [45]. The importance of LDHB in the progression of various cancers has been reported [46]. The direct interaction of Aurora-A with LDHB results in phosphorylation of the enzyme (S162). The resulting increases in catalytic activity boost NAD^+^ regeneration, glycolytic flux and the biosynthesis of glycolytic metabolites, all of which facilitate tumor progression [47].

Together, LDHA and LDHB not only ensure the metabolic plasticity of neoplastic cells—which allows them to adapt to hostile environmental or cellular conditions, such as the increased production of reactive oxygen species—but also regulate cell death (apoptosis and autophagy) [48]. Thus, the role of these enzymes in tumor biology is more complex than was initially expected, but this may also create an opportunity to target them in the treatment of cancer. 

## 4. Lactate Transport

The metabolic processes of cancer cells result in the accumulation of lactate in the cytosol, which together with the accumulated H+ must be released into the extracellular space: (i) to prevent intracellular acidification and (ii) to support high rates of glycolysis, as high concentrations of lactate in the cytosol reduce the glycolytic rate by inhibiting the rate-limiting enzyme PFK-1 [49,50]. The release of lactate and H+ into the TME [15] is mediated by monocarboxylate transporters (MCTs) and Na-driven proton extrusion, respectively [51,52]. The reliance of lactate removal on MCTs is necessitated by the fact that lactic acid is hydrophilic and a weak acid and thus unable to freely diffuse across the cell membrane. The pKa of the lactate/lactic acid pair is 3.8 at physiological pH, with lactic acid immediately dissociating into lactate (base form) and hydrogen (H+). Nonetheless, several studies conducted in the 1990s clearly showed that lactate is not responsible for extracellular acidosis; rather, ATP hydrolysis coupled with glycolysis is the main source of H+ production that leads to a decrease in the extracellular pH [35].

MCTs belong to the solute carrier transporter family (SLC), composed of 52 families of membrane transport proteins. One of these families, SLC16, encodes 14 MCT isoforms with significant roles in the absorption, tissue distribution and clearance of both endogenous and exogenous compounds [53]. However, only four isoforms (MCT1–4) are lactate transporters, although they can also carry other monocarboxylates, such as pyruvate and ketone bodies [54]. MCT1 facilitates both lactate and pyruvate upload, whilst MCT4 exports lactate and H+, and thereby contributes to the reversed pH gradient [55]. The direction of lactic acid exchange depends on the concentration gradients of both protons and monocarboxylate ions. Thus, MCT4 [56] serves to gate the export of lactate, while MCT1 can facilitate both its import and export depending on the pH gradient [57]. 

In doing so they promote a metabolic interchange between glycolytic and oxidative tumor cells [58], as the lactate released by cancer cells is used by neighboring stromal cells as an energy source [59]. For example, endothelial cells use lactate both as a signaling molecule and as a metabolic intermediate in neo-angiogenesis [60]. Differences in the regulation of the expression and activity of lactate transporters support the intercellular lactate exchange model shown in Figure 2.

## 5. The Lactate Receptor GPR81 and other Modes of Lactate Transport

Lactate also exerts its pro-tumoral activity independent of MCTs, by binding to its receptor, GPR81, at the cell surface [56]. Lactate signaling via GPR81 does not require H^+^ or MCTs, nor is a conversion to pyruvate or an alteration in the intracellular NADH/ NAD^+^ ratio a prerequisite.

GPR81 belongs to the hydroxy carboxylic acid receptor (HCAR) subfamily, which is composed of three members: HCAR1 (GPR81), HCAR2 (GPR109A) and HCAR3 (GPR109B) [62]. It was discovered by Ge et al. in 2008 [63] and was first identified in the plasma membranes of adipocytes. GPR81 acts synergistically with insulin to decrease intracellular cAMP levels and lipolysis in the fed state, with a clear link to obesity [64]. L-lactate is the receptor’s natural ligand and has an EC50 of ≈5 mM. Recently, the lactate-GPR81 system has been implicated as a signaling mechanism in neuroprotection, angiogenesis and the regulation of neuronal firing [65,66,67]. A role for lactate via GPR81 binding has also been observed in infection, inflammation and the suppression of innate immune cell function [68,69]. 

The expression of GPR81 on the membranes of cancer cells was first reported in 2014, in a study involving different cancer cell lines and tumor samples resected from pancreatic cancer patients [70]. Lactate signaling via GPR81 is especially unique from everything we have seen until now. The signaling through GPR81 does not require H^+^ or MCT-driven import, nor does it require a conversion to pyruvate or alterations in the intracellular NADH/ NAD^+^ ratio. Despite its negligible expression in benign cells, the receptor was shown to be upregulated in malignant cells derived from the same tissues. The 2014 study of Roland et al. [70] demonstrated that the silencing of GPR81 in pancreatic cancer cell cultures with high glycolytic ratios and high lactate concentrations in the media resulted in significantly reduced mitochondrial activity and obvious increases in cell death. The importance of GPR81 as a pro-tumorigenic element has also been determined in breast, hepatocellular and lung cancers and cervical squamous-cell carcinoma [71,72,73]. Recently, Xie et al. [74] reported that lactate itself induces the expression of GPR81 in cancer cells via transcriptional activation involving the Snail/EZH2/STAT3 transcriptional complex.

GPR81 may contribute substantially to tumor growth and proliferation by responding to lactate in the extracellular environment in an autocrine or a paracrine manner. In the former, lactate released by the tumor cells is taken up by GPR81 expressed on those cells. Alternatively, extracellular lactate may act on GPR81 expressed on the non-cancer cells located in the TME that support tumor growth [75], including immune cells (dendritic cells and macrophages), adipocytes and endothelial cells [69,76,77]. The presence of a high concentration of lactate in the extracellular environment of the tumor is associated with a poor prognosis [78]; thus, GPR81 signaling, whether autocrine or paracrine, is likely to be involved in the promotion of tumor growth and/or metastasis. 

Finally, an alternative route for venting lactate anions from poorly perfused cells involves gap junctions assembled from connexin proteins. They form intercellular channels that couple the cytoplasms of neighboring cells into a syncytium and establish high conductance pathways for the passage of small molecules [79]. Several reports produced evidence that connexins can, in fact, facilitate late-stage disease in certain cancers [80]. In spheroids of PDACs (pancreatic ductal adenocarcinoma cells), researchers were able to demonstrate that connexin 43 (Cx43) channels are important conduits for dissipating lactate anions from glycolytic PDACs. Furthermore, lactate entry into the better-perfused recipient cells has a favorable alkalinizing effect and supplies substrate for oxidative phosphorylation. Cx43 is thus a novel target for influencing metabolite handling in junctional-coupled tumors [81].

The different modes and activities of the lactate transporters discussed herein are summarized in Figure 2.

## 6. Lactate Exchange

Solid tumors are typically nutrient-starved, as their rapid growth drives an increased rate of nutrient consumption further worsened by a deficient tumor vascular supply. To bypass these limitations, tumors have developed various nutrient-scavenging strategies, including lactate exchange [82]. Figure 2 depicts the exchange of lactate between tumor cells, stromal cells and the TME.

Most tumors are heterogeneous, made up of oxidative cells, glycolytic cells and stromal cells. Their complex metabolic relationship includes the exchange of lactate in the form of metabolic symbiosis [83]. Oxidative tumor cells are located close to tumor blood vessels and predominantly oxidize lactate to pyruvate in a reaction catalyzed by LDHB, with the simultaneous output of NADH and H^+^. In addition to their favorable location allowing a high nutrient availability, these cells express MCT1 and are thus able to take up the lactate secreted into the TME by glycolytic tumor cells and stromal cells through their expression of MCT4 [17,59]. Because oxidative tumor cells preferentially use lactate as an oxidative fuel, they spare glucose such that it becomes available for both glycolytic tumor cells and stromal cells, via anaerobic and aerobic glycolysis, respectively [83]. Both glucose transporter-1 (GLUT-1) allowing cellular glucose uptake and MCT4 are induced in the distal hypoxic cells of a tumor, with a clear dependency on HIF-1-α [83].

## 7. Lactate and Cancer-Related Genes

A “triad” of transcription factors consisting of HIF-1α, c-MYC and p53 are largely responsible for the glycolytic phenotype in cancer [84]. In particular, the up-regulation of HIF-1α and c-MYC together with the suppression of p53 induce a metabolic switch to glycolysis in cancer cells by inducing the overexpression of glycolytic enzyme by 2 to 500-fold [85]. Conversely, lactate, as an active metabolite of aerobic glycolysis and glutaminolysis, alters the transcriptional activity of several key oncogenes and other driver genes involved in metabolic reprograming, cell cycle regulation and proliferation [26]. San-Millan et al. 2020 [86] demonstrated that lactate acts as an oncometabolite in the MCF7 human breast cancer cell line, as it increases the transcriptional activity of MYC and that of PIK3CA, AKT1, HIF-1α and BRCA1, all of which contribute to an upregulation of the glycolytic pathway in this cancer cell line [87,88].

Among the oncogenes activated by lactate is MYC, a potent mediator of tumorigenesis whose deregulation has been found in a variety of cancers [89]. MYC participates in glucose metabolism by increasing the expression of the glucose transporter GLUT1 and by upregulating the expression of glycolytic enzymes, including hexokinase 2 (HK2), PFK-M1 and enolase 1 (ENO1) [90,91]. MYC also enhances expression of the M2 isoform of pyruvate kinase (PKM2), an enzyme essential for aerobic glycolysis and present in virtually all tumors [92], by promoting the expression of hnRNP splicing factors, as demonstrated in glioma [93]. 

During glycolysis, NAD^+^ is reduced to NADH. The regeneration of NAD^+^ and thus the maintenance of the glycolytic flux depends on LDHA, which converts the pyruvate derived from both the glycolytic and the glutaminolytic pathway into lactate [94]. The overexpression of MYC enhances LDHA expression, with the increased production of lactate then leading to extracellular acidification, as discussed above [95]. MYC also promotes lactate secretion, by enhancing the expression of MCT1 [5], and the uptake of glucose indirectly, by blocking the transcriptional function of MondoA, which in turn inhibits the thioredoxin-interacting protein (TXNIP), a negative regulator of glycolysis [96]. 

HIF1-α, which is also overexpressed in tumor cells, increases the transcription of genes regulating glucose transport and glycolytic enzymes [97]. Moreover, it causes a metabolic reprogramming that leads to the Warburg effect and thus to lactate production. HIF1-α has been implicated in breast cancer tumor growth and metastasis, and in tumor aggressiveness associated with a poor prognosis [86,98].

Mutations in the Ras oncogene are found in many types of human cancers and drive the metabolic phenotype of cancer cells toward aerobic glycolysis [99]. Ras activates the mammalian target of rapamycin (mTOR) via the PI3K-Akt-mTOR pathway, which promotes glycolysis by inducing HIFs [100,101].

Tumor suppressor p53 is a transcription factor that regulates diverse biological functions, including cellular energy metabolism. It plays a pivotal role in balancing glycolysis, which it inhibits, and oxphos [102], which it promotes [102], including by directly regulating LDHA expression at the transcriptional level. As such, the dysregulation of p53 is an important driver of the metabolic switch to glycolysis in cancer cells. Among the other critical functions of p53 are the control of cell proliferation, invasion and the induction of apoptosis [103]. As a potent negative regulator of HIF-1α, it blocks the latter’s accumulation in normoxia and hypoxia [104] and induces inhibitory microRNA-107 [105]. 

Other transcription factors have also been implicated in the role of lactate in tumor metabolism. For example, in 2018, Li et al. [106] showed that the transcription factor sine oculis homeobox 1 (SIX1) can regulate the Warburg effect by binding to promoters and recruiting the histone acetyltransferases HBO1 and AIB1. These enzymes in turn induce the expression of LDHA and many other glycolytic genes (GLUT1, HK2, PFKL, ALDOA, GAPDH, PGK1, ENO1, PKM2) involved in the glycolysis pathway. 

TWIST1, a transcription factor and master regulator of the epithelial-to-mesenchymal transition (EMT), promotes the invasion and metastasis of cancer cells. In pancreatic ductal adenocarcinoma, TWIST1 directly increases the transcription of several glycolytic genes, including GLUT1, HK2, ENO1 and PKM2. Transcriptional regulation by TWIST1 is not dependent on HIF1α or c-Myc [107]. 

In a recent study, Zhang et al. [108], showed that the “lactylation” of histone lysine residues serves as an epigenetic modification that directly stimulates gene transcription in human and mouse cells. Among their novel findings was a dose-dependent increase in lysine lactylation (Kla) in response to exogenous L-lactate and that endogenous production of lactate is a key determinant of histone Kla levels. Bhagat et al. [109] also recently demonstrated that tumor-mediated lactate can elicit epigenomic reprograming—proven in pancreatic ductal adenocarcinoma. However, the influence of lactate on the cancerous epigenome is thus far poorly understood.

## 8. Lactate as a Key Molecule in the “Immune Scape”

The TME, through the actions of its stromal cells, is in a state of constant modification as the tumor progresses. As noted above, CAFs, tumor endothelial cells and immune cells make up the cellular components of the TME. On the one hand, innate (macrophages, neutrophils, dendritic cells, innate lymphoid cells, myeloid-derived suppressor cells and natural killer cells) and adaptive (T and B cells) immune cells present in the TME are responsible for the detection and elimination of cancer cells [110]. On the other, the ability of tumor cells to secrete anti-inflammatory cytokines allows the recruitment of immunosuppressive cell populations to the TME, where they directly inhibit immune responses (Figure 3) [111,112].

In the tumor microenvironment some areas can reach up to 40 mM lactate concentrations, which tumor cells can cope with [20]. The question is, how do infiltrating immune cells handle a lactate-rich microenvironment?

There is evidence of a harmful effect of high concentrations of lactate in the TME, including the tumor-infiltrating immune cells, but, paradoxically, immune cells contribute to intratumoral lactate production [113]. However, this contribution is relatively modest, as it depends on the number of immune cells recruited, their differentiation/activation statuses and whether they have become dysfunctional due to the immunosuppressive mechanisms of the tumor. 

Lactate contributes to the immune escape of tumor cells by inducing the apoptosis of natural killer (NK) and natural killer T (NKT) cells, both of which exhibit antitumoral activity [114,115]. 

Lactate concentrations > 20 mM were shown to induce the apoptosis of both cell types, which may explain their smaller proportions in tumors with higher concentrations of lactate [116]. Lactate also blocks interferon (IFN)-γ and interleukin (IL)-4 production by antitumoral NKT cells in the TME, via the inhibition of mTOR signaling, thereby preventing the activation of these immune cells [115]. Fisher et al. [21] showed that lactate inhibits T-cell proliferation and alters cytokine production through cultured cytolytic T lymphocytes. Lactate/Gpr81-induced immunosuppression also inhibits host defense against tumor growth [71,90].

Dendritic cells (DC) are antigen-presenting cells that play a major role in immune responses. One of the main functions of DCs is the recognition of tumor cells, by processing and presenting tumor antigens through the MHC-II and MHC-I, leading in turn to the activation of CD4^+^ and CD8^+^ T lymphocytes, respectively, in response to environmental cytokines such as IL-12, TNF-α and IFN-γ produced by NK and NKT cells and macrophages [117]. Lactate prevents DC differentiation and causes the cells to become tolerogenic, leading to an increase in the production of IL-10, a potent immuno-suppressive cytokine [118] that inhibits the production of proinflammatory cytokines such as IFNγ, TNFα, IL-1β and IL-6; moreover, IL-10 prevents DC maturation and T cell activation [119].

Lactate also promotes the development of myeloid-derived suppressor cells (MDSCs), a heterogeneous population of immature myeloid cells that are able suppress both innate and adaptive immunity by preventing the maturation of dendritic cells (DCs); suppressing natural killer (NK) cell cytotoxicity; inhibiting T cell activation; and favoring the differentiation of regulatory T cells and thus disturbing innate and adaptive immune responses [120].

In many cancers, tumor-associated macrophages (TAMs) are constantly recruited to the tumor environment by the CCL2 chemokine, which attracts CCR2+ monocytes circulating in the blood. Following their arrival in the TME and subsequent differentiation, the resulting TAMs contribute to neoplastic growth, invasion and metastatic diffusion by translating instructive signals delivered by transformed cells. Lactate in the TME is taken up by TAMs through their MCTs, which are located on the cell membranes. Among the lactate-induced responses are the HIF-α-induced transcriptions of vascular endothelial growth factor (VEGF) and the metabolizing enzyme arginase-1 (ARG1), which promote TAM polarization [121]. The latter results in the expression of a set of genes that are common to M2-type macrophages, a specialized subset of TAMs that mediate in inflammation resolution and tissue remodeling. In a breast cancer model [122], lactate was shown to activate the ERK/STAT3 signaling pathway and thereby TAM polarization and M2 macrophage differentiation, which in turn promoted tumor proliferation, migration and angiogenesis.

Lactate is also involved in tumor evasion of the immune response, including via its receptor GPR81. In co-cultures of GPR81-expressing lung cancer cells and Jurkat T cells, both cell proliferation and IFN-γ production by the latter were reduced compared to co-cultures with lung cancer cells lacking GPR81 [73]. Activation of the GPR81 receptor on the surface of lung cancer cells upregulates the membrane expression of PD-L1 to block immune responses to the tumor [86]. Similarly, the blockage of LDHA in tumor cells improves the efficacy of anti-programmed cell death-1 (PD1) therapy [116]. Brown et al. [123] showed that antigen-presenting DCs express GPR81, whose lactate-mediated activation suppresses the cell-surface expression of MHC-II, thereby compromising the ability of DCs to present tumor-cell antigens to T cells [123]. 

Taken together, these studies provide multiple lines of evidence that lactate is an important component of the TME, promoting tumor growth and immunosuppression and therefore carcinogenesis. 

## 9. Lactate in Tumor Metastasis 

As tumor cells proliferate, room must be made for their expansion, which requires degradation of the ECM, the invasion of local tissues and other processes. The acidification of the extracellular space that occurs following the transport of lactate to the TME modifies the binding properties of tumor cell surface integrins, improving their binding to ECM components and the subsequent migration of tumor cells [9,124]. Acidification also activates tumor-cell proteinases, such as matrix metalloproteinases-9 [125], cathepsin B and hyaluronidase-2, which degrade the surrounding matrix and promote tumor cell migration [126]. A decreased extracellular pH increases the density and length of tumor cell “invadopodia,” responsible for the movements that support tumor cell invasion [127]. 

Evidence from mouse models of cancer suggests that neutralizing the external acidity of the tumor with oral buffers is an effective strategy for the prevention and inhibition of metastasis [128,129,130]. In fact, a recent study showed the association of combined chemotherapy and alkalization therapy with more favorable outcomes in patients with advanced and recurrent pancreatic cancer who had increased urine pH after alkalization therapy [131]. A pilot phase I clinical study by a research group from the University of Arizona examined the safety of the long-term consumption of sodium bicarbonate (ClinicalTrials.gov identifier: NCT02531919), but detailed results have yet to be reported. 

A correlation between high concentrations of lactate in the TME and a greater propensity of tumors to metastasize has been reported [132,133]. Indeed, in different forms of human cancers the strong correlation between lactate levels and metastasis is well-established, including cervical cancer [134], head and neck cancer [20,135], colorectal adenocarcinoma [136] and gastric cancer [137]. For example, in cervical cancer a high lactate concentration correlates with poorer overall and disease-free survival [19].

Nonetheless, the mechanism underlying lactate’s involvement in metastasis is not fully understood. In glioma cells, lactate induces the expression of transforming growth factor-β2 (TGFβ2), a key regulator of glioma cell migration [138]. The addition of exogenous lactate to cultures of different cancer cell lines increases cell motility and random migration in a concentration-dependent manner [139]. In cells exposed to lactate, MCT1 mRNA and protein expression increase rapidly [140]. Clinical studies have shown that high-level MCT1 expression is associated with invasion in different cancers, including MCT1 in non-small-cell lung cancer [141] and MCT4 in melanoma [142]. Zhao et al. [143] proposed a decrease in NF-κB signaling coupled with MCT1 repression as a molecular mechanism to decrease osteosarcoma cell migration. A similar conclusion was reached in a study of cervix and breast cancer cells [144]. However, the pro-migratory activity of MCT1 is independent of its transporter activity, and the direct or indirect interaction of MCT1 with upstream components of the NF-κB signaling pathway may support its activity. Like the knockdown of MCT1, MCT4 knockdown seems to impair the migration and invasion of different cell lines [128,145]; Gallagher, S.M. et al. were able to demonstrate that MCT4 directly interacts with β1-integrin at the lamellipodium of migrating cells [146]. Since integrin conformation is pH-sensitive [147], the loss of MCT4 activity may locally modify the transmembrane pH gradient and therefore integrin signaling and cell adhesion.

Furthermore, CD147, a chaperone protein shared by MCT1 and MCT4, triggers cancer cell migration, invasion and metastasis, specifically through activation of matrix metalloproteinases (MMPs) [148]. As MCT1 and CD147, and MCT4 and CD147 are mutually stabilizing at the cell plasma membrane (Figure 2) [149], silencing of MCT1 or MCT4 might impair CD147 expression and function.

In their study of a murine model of breast tumors, Rizwan et al. [150] demonstrated that both LDHA expression and overall lactate production correlated with disease severity. The knockdown of LDHA was shown to delay metastasis and increase overall survival [150]. A link between LDHA/LDH5 overexpression and the epithelial-mesenchymal transition that characterizes metastatic disease was also recently reported [151].

The strong correlation of the lactate level in the TME with metastasis in different forms of human cancers, such as cervical cancer [134], head and neck cancer [20,135], colorectal adenocarcinoma [136] and gastric cancer, is well-established [137]. In cervical cancer, a higher lactate concentration correlates with poorer overall survival and a poorer disease-free survival [19].

## 10. Lactate in Therapy Resistance

The tumors of patients treated with antineoplastic agents frequently become drug resistant, in the form of either reduced responsiveness (primary resistance) or tumor relapse and progression (secondary resistance) [152]. Resistance has been attributed to cell-autonomous and non-cell autonomous mechanisms [153]. The TME has been implicated in the latter through various mechanisms, including hypoxia, extracellular acidity and lactate production [154]. The acidic TME seems to create a chemical barrier which means the extracellular accumulation of some chemotherapeutic drugs that usually enter the cells via passive diffusion, which limits their effects and activity. [155].

“Ion trapping” is a biological process that regulates passive permeability through the cellular membrane of charged compounds [156]. As a result, drugs can be impeded from reaching their targets because they get trapped on the wrong side of the cellular compartment. Both the intrinsic pKa values and the pH of the solution define the ionization of a molecule. Many small cancer drugs (weak bases or acids) are ionizable and therefore prone to ion trapping resistance [157]. Doxorubicin, a weakly basic anticancer drug (neutral in acidic conditions), can freely pass the membrane and enter the cell; it is then reduced and the ionized majority of the drug molecules are trapped on the extracellular side. The lack of efficacy of doxorubicin is associated with the ion trapping and low tumor tissue distribution [158]. On the other hand, an acidic extracellular environment favors the permeability of weakly acidic drugs and their cytotoxic activity. Anticancer drugs such as camptothecin, chlorambucil and melphalan are known to increase by extracellular acidosis. However, methotrexate, a weakly acidic drug, shows decreased cytotoxic activity under acidic conditions [159] due to breast cancer resistance protein (BCRP) that transports methotrexate, and in acidic conditions the electrostatic interaction between methotrexate and BCRP increases, which mediates drug efflux and consequently multidrug resistance [160]. Following the binding of lactate in the TME to MCT1, intracellular signaling pathways are activated that alter the expression of downstream effector molecules and allow tumor cells to become drug-resistant, via the AKT/mTOR, NF-κB and STAT3 signaling pathways. For example, activation of the mTOR pathway initiates metabolic symbiosis in cancer cells which thus become resistant to VEGF inhibitors [161]. The mechanism involves a switch by the tumor cells to the senescence-associated secretory phenotype (SASP), which confers non-cell-autonomous resistance. NF-κB signaling controls the expression of immunomodulatory and secretory factors, such as IL-6 and IL-8, which modulate the initiation and persistence of the SASP. In lymphoma, this phenotype is destroyed by NF-κB inhibition, leading to an escape from immunosurveillance by NK cells and p53 inactivation, and therefore to drug resistance [162]. In the TME, macrophages, neutrophils and CAFs are the major cell types that secrete IL-6 and IL-1β; they are also responsible for STAT3 activation in tumor cells [163]. The activation of STAT3 and downstream effectors may confer drug resistance by initiating the EMT, suppressing epigenetic tumor suppressor miRNAs and enhancing the expression of antiapoptotic proteins [164,165]. Moreover, STAT3 expression in tumor cells may also enhance the expression of Rab family proteins, which facilitates exosome release to confer cisplatin resistance in ovarian cancer [166]. Acidic microenvironment is also involved in the increased rate of endosomal-lysosomal trafficking, and the increased release of extracellular vesicles (EVs) by tumor cells [167]. Thus, it is reasonable to think that cancer cells under the stress of the overly acidic microenvironment need to remove their toxic byproducts, and one of the mechanisms at their disposal is exosome elimination. Chemotherapeutic drugs such as cisplatin have been demonstrated to be eliminated via exosomes which thus participate in chemorresistance [168].

Tumor-derived exosomes have been detected in a wide variety of cancers and may also play significant roles in carcinogenesis and metastasis [169]. In cancer patients, a higher number of secreted exosomes correlates with a poor prognosis [170].

In in vitro models using NSCLC cell lines, Apicella et al. [171] showed that lactate is a key mediator of tumor cell resistance to therapy based on tyrosine kinase inhibitors (TKIs), particularly JNJ-605, a c-MET receptor tyrosine kinase inhibitor, and erlotinib, an epidermal growth factor receptor (EGFR) inhibitor [171]. Another study examined the metabolic inhibitors BEZ235, LY294002 and GDC0942 (as PI3K inhibitors) and GDC0980 (a dual PI3K/mTOR inhibitor) and was able to demonstrate the inhibition of breast cancer cell proliferation in high-glucose medium [172]. However, when lactate was used as the primary substrate, the cells were completely resistant to the inhibitors, suggesting that cancer cells able to rely on glycolysis by utilizing lactate are less sensitive to PI3K/mTOR inhibitors.

Lactate has also been implicated in radiotherapy resistance. In nude mice xenografted with human head and neck cutaneous squamous cell carcinoma cell lines and then treated with irradiation (4Gy) within 6 weeks, a high lactate concentration correlated with radioresistance [173].

## 11. Therapeutic Strategies Targeting Lactate

The findings discussed above provide strong evidence of a dual role for lactate in tumors, as a metabolic fuel and as a signaling molecule, thereby positioning lactate at the intersection of cancer initiation and progression. Targeting the aberrant lactate homeostasis of tumor cells offers a promising approach to cancer therapeutics, since any interference in the expression and/or functioning of the molecules that contribute to the deregulated glucose and/or glutamine metabolism will inevitably impact lactate production and release [174]. The intercellular exchange of lactate between oxidative and glycolytic tumor cells or tumor cells and stromal cells, (including endothelial cells and fibroblasts) can be targeted as well. The development of LDH and MCT inhibitory strategies may thus be promising avenues of research. In the following, we highlight recent findings obtained using available molecules directed at interfering with lactate production and transport, with a focus on those involving LDHs and MCTs.

### 11.1. Targeting Lactate Production

An effective therapeutic approach may be targeting LDHs (Table 1) that mediate the bidirectional conversion of pyruvate into lactate. Because LDHA is the predominant isoform expressed in glycolytic tumors, several LDHA-targeting compounds have been investigated. Among these, the ability of FX-11, a gossypol derivative (AT-101) [175]; galloflavin [176]; and N-hydroxyindole-based compounds [177] to preferentially inhibit LDHA has been demonstrated [178]. 

In tumor xenograft models, FX-11 efficiently inhibited the growth of P493 and P198 pancreatic cancer cells. In an alternative approach using a human lymphoma xenograft model, FK866, which hampers NAD^+^ synthesis, was tested both alone and in combination with FX-11, and potently inhibited lymphoid cell proliferation [175]. These results provide strong evidence that LDHA is necessary for tumor progression [175]. 

Gossypol, also known as AT-101, is a nonselective inhibitor of LDH, whose antitumor activity has been attributed to its additional ability to inhibit the activities of anti-apoptotic Bcl-2 protein family members. It has been tested in several phase I and phase II clinical trials (Table 1), either as monotherapy or in combination with chemotherapy in several tumor types, but in the majority of studies the response rates were insignificant [179,180].

Heat shock transcription factor 1 (HSF-1) regulates the expression of heat shock proteins (HSPs), which are essential for cell survival, and the heat shock response (HSR). In addition, HSF-1 regulates glucose metabolism by activating the expression of LDHA [181]. Galloflavin and oxamate, another inhibitor of LDH activity that directly competes with its natural substrate, were tested in a model of hepatocellular carcinoma (Table 1). The results indicated that LDH inhibition is an efficient way to dampen a constitutively activated HSR in cancer cells, by hindering the functions of the three major molecular chaperones (HSP-90, HSP-72 and HSP-27) involved in tumorigenesis. Furthermore, both compounds resulted in cell senescence [174]. However, oxamate has never been used in clinical trials because its activity requires concentrations in the millimolar range [182,183].

Other potent inhibitors of human LDH include 2-thio-6-oxo-1,6-dihydropyrimidine, with effective cellular in vitro cytotoxicity in pancreatic carcinoma cells (MIA PaCa-2 cell line) and in a mouse model of cancer [184]. High-throughput small-molecule screening using a library containing ≈2 million compounds was conducted to identify small-molecule inhibitors of LDHA. One such inhibitor, GNE-140, efficiently inhibited murine B16 melanoma and human adenocarcinoma and pancreatic carcinoma cells in vitro (Table 1). The drug’s activity was dependent on the metabolic activity of the cells [185,186]. However, to the best of our knowledge, no clinical trials of LHD small-molecule inhibitors have been registered to date.

Unfortunately, none of the above-discussed compounds nor pyrazole-based inhibitors of LDH [187] have progressed to the point of being clinically viable forms of treatment. Given the importance of lactate metabolism in different types of cancers, optimizing existing compounds while continuing the search for and development of new LDHA inhibitors would be a reasonable strategy.

A very promising novel compound (compound 11) was reported by Fang A et al. [188], who used docking-based virtual screening and biological assays. When tested in a MG-63 osteosarcoma cell line, compound 11 inhibited LDHA and induced apoptosis by decreasing lactate formation and extracellular acidification [188]. Nevertheless, further experiments with different types of cancers are needed to ensure the biological efficacy of this drug.

Recently, Kim et al. [189] identified several promising selenobenzene compounds with LDHA-inhibitory activity. The most potent was 1-(phenylseleno)-4(trifluoromethyl) benzene (PSTMB), which inhibited cell proliferation and induced apoptosis in several human cancer cell lines, including lung cancer (NCI-H460), breast cancer (MCF-7), hepatocellular carcinoma (Hep3B), malignant melanoma (A375) and colorectal adenocarcinoma (HT29) lines. PSTMB reduces both LDHA activity and lactate production under normoxic and hypoxic conditions, by inhibiting enzyme activity directly, rather than enzyme expression [189] (Table 1).

Zhou et al. [182] characterized a compound referred to as 24c, a novel potent LDHA inhibitor that interacts directly with the enzyme’s binding pocket. Compound 24c was obtained by a hit-to-lead optimization using an in-house library. In the MiaPaCa-2 pancreatic carcinoma cell line, compound 24c resulted in dose-dependent reductions of cell growth and cell cycle arrest and apoptosis (Table 1). Furthermore, it suppressed tumor growth in a xenograft model [190]. These results suggest the use of compound 24c as a lead compound in the development of new, more potent LDHA inhibitors [190].

With the aid of in silico methods, Jafary F et al. [191] designed novel peptides that interfere with LDHA activity, by anchoring the enzyme’s subunits such that tetramerization and therefore activity are blocked [191]. However, these peptides must be developed and then tested in in vitro before their biological action in vivo can be evaluated.

### 11.2. Targeting Lactate Transporters

Targeting MCTs is likely to have dramatic effects on intercellular lactate exchange (Figure 2). MCT1-specific inhibition damages lactate influx, forcing a cellular metabolic switch from the lactate that fuels oxphos to aerobic glycolysis, thereby indirectly causing the death of hypoxic cancer cells due to glucose deprivation [61,83]. MCT1 targeting can affect the intercellular lactate exchange, which is very important for cancer cell adaptation to glucose depletion [61,83]. In addition, the targeting of MCT1 weakens tumor resistance to anti-angiogenic therapy [192].

MCT inhibitors include AR-C155858 [193], which non-specifically targets MCT1 and MCT2, and SR13800, whose target is MCT1 [194]. In addition, promising preclinical success has been obtained with the AstraZeneca compound AZ3965, an inhibitor of both MCT1 and MCT2 but with 6-fold greater selectivity for MCT1 [195]. 

The in vitro anti-tumor activity of AZ3965 is mediated by a significant impairment of lactate production leading to massive tumor cell die-off [196]. AZ3965 was shown to be effective when tested in models of Burkitt’s lymphoma, and breast, gastric and small-cell lung cancer [194,195] and is currently the focus of a phase I/II clinical trial (ClinicalTrials. Gov NCT01791595) (Table 2). A possible drawback is that data from preclinical and retrospective analyses suggest that when MCT1 is inhibited, MCT4 is able to compensate for its function. However, Ždralevic, M. et al. [197] demonstrated that this mechanism also offers metabolic vulnerabilities for therapeutic interventions, via “ferroptosis”-induced cell death [197]. Intriguingly, in the only study examining the effect of AZD3965 on angiogenesis, the drug did not alter the vascularization of a small cell lung carcinoma xenograft [198].

In murine models of cancer, administration of the MCT inhibitor α-cyano-4- hydroxycinnamate (CHC) decreased tumor growth by inducing necrosis in the tumor core, associated with the extinction of hypoxic tumor areas [83]. The authors proposed that while oxidative cancer cells adapt to MCT1 inhibition by switching to alternative substrates, glycolytic cancer cells cannot depend on lactate exchange for survival [83]. However, MCT1 inhibition interferes with other oxidative cancer cells, such as stromal cells, that take up additional lactate in the TME. While co-cultures of tumor cells and CAFs fuel cancer cell proliferation, either CHC administration or MCT1 knockdown is able to disrupt this relationship, hence impairing cancer cell proliferation [199]. Moreover, MCT1 targeting in endothelial and cancer cells, using a silencing approach or CHC administration, was shown to disrupt lactate-induced angiogenesis, both in vitro and in murine models of cancer in vivo [60,200] (Table 2).

Another novel selective MCT1 inhibitor is BAY-8002, which is six times more selective for MCT1 than for MCT2, has no activity against MCT4 and has no off-target effects [201]. 

Competition studies demonstrated the similar mechanisms of action of BAY-8002 and AZD3965, based on their mutual displacement. Moreover, cancer cells with inhibited MCT1 activity increase their levels of oxidative mitochondrial metabolism and become more sensitive to ETC complex I inhibitors, such as metformin, phenformin and BAY87-2243 [196,202,203], and to the GLS1 inhibitor bis-2-(5- phenylacetamido-1,3,4-thiadiazol-2-yl) ethyl sulfide (BPTES) (Table 2) [204]. Accordingly, the ability of MCT1 to transport additional exogenous compounds should be kept in mind, as inhibition of the enzyme may reduce/augment the toxic effects of these molecules [205]. In the development of combination treatments, the potential involvement of MCT1 in multidrug resistance should be taken into account [206].

MCT4 mediates lactic acid efflux from glycolytic cancer cells and is therefore an important pH regulator [207]. The inhibition of MCT4 would acidify the cytosol of glycolytic cancer cells and thereby induce their death. MCT4 is highly expressed in many tumors, particularly in hypoxic regions of the fast-growing tumor mass, due to its HIF1-dependent expression. The disruption of lactate exchange between different cell populations by MCT4 knockdown may also offer an effective therapeutic strategy. 

Among the selective inhibitors of MCT4 are diclofenac [208] and bindarit (2-[(1-benzyl1H-indazol-3-yl) methoxy]-2-methylpropanoic acid) [209] (Table 2). The AstraZeneca MCT4 inhibitor AZ93 reduces the proliferation of various cancer cell lines in which MCT1 has been inhibited as well [210]. Indeed, it may be that only the concurrent inhibition of MCT1 and MCT4 can impair tumor growth, especially under hypoxic conditions. In this context, syrosingopine, a dual inhibitor of MCT1 and MCT4, showed potential antitumor benefits in vivo [189]. Lonidamine, another dual inhibitor, was particularly effective in sensitizing tumors to other therapies [196,211] (Table 2). For example, additive or synergetic effects have been observed following MCT1, MCT2 or MCT4 inhibition in combination with chemotherapy [71,143,212] and radiotherapy [71]. 

Finally, MCT1 and MCT4 localization and maintenance at the plasma membrane are influenced by CD147/basigin, a member of the co-chaperone immunoglobulin-family. Targeting CD147 may therefore offer a novel strategy to inhibit the activity of both transporters. Among the agents tested thus far are an organomercurial compound, p-chloromercuribenzene sulfonate (pCMBS), which disrupts MCT association with CD147 [213]; AC-73, which targets CD147 dimerization [214]; and humanized anti-CD147 antibodies [215] (Table 2). However, CD147 is expressed in other tissues and can act as co-chaperone for other membrane proteins, such that its safety as an anti-cancer target must be carefully evaluated.

## 12. Conclusions and Remarks

In this review, we present an integrated assessment of the role of lactate in tumor development and growth. The Warburg effect and other alterations in tumor metabolism have been recognized as hallmarks of cancer for several decades, but the wide-ranging roles of lactate and acidosis in tumorigenesis have only recently been recognized.

Glycolytic cancer cells increase their uptake of glucose and nutrients and their production of lactate even under aerobic conditions. They are also able to adapt to hypoxic and low-nutrient microenvironments and engage in lactate exchange with the oxidative cancer cells adjacent to blood vessels and sustained by the high nutritional availability offered by this location. These cells are essential to rapid tumor progression. Our review also explored both lactate’s ability to stimulate angiogenesis and the effects of lactate on immunosuppression and other immune cell functions.

Renewed interest in cancer metabolism during the last 15 years has alerted researchers to the potential of targeting tumor metabolism in the treatment of cancer. However, thus far, metabolic targeting approaches have been effective in the preclinical setting, but their translational impact remains limited. This may be a consequence of the metabolic heterogeneity of the cell populations that constitute the tumor bulk. Even in seemingly identical tumor tissues, the metabolic behavior of individual tumor cells differs, resulting in a complex metabolic mosaicism, with some cancer cells driven by oxphos and others by glycolysis. This preference may be caused by genetic heterogeneity of the tumor, low vs. high-nutrient perfusion, hypoxic vs. non-hypoxic exposure and/or by the consequences of a highly stroma-infiltrated tumor mass. The aggressiveness of tumors is derived in part from their higher metabolic plasticity, which no doubt has contributed to the failure of previous trials with antimetabolic drugs in single-agent administrations. 

Driving tumor and stromal cells to adopt a more homogeneous metabolic phenotype through combinations of two or more drugs may thus be more successful. For example, anti-angiogenic drugs may generate a more hypoxic TME and therefore a more homogeneous metabolic phenotype. Targeting lactate metabolism in combination with immunotherapy to enhance the efficacy of the latter also holds promise. Using a murine melanoma model, Daneshmandi S. et al. [116] demonstrated that the blockage of LDHA increased the number and cytolytic activity of NK cells and cytotoxic T lymphocytes, resulting in reduced tumor growth, when initiated in combination with anti-PD-1 therapy [116].

Further studies of the metabolic pathways of tumor cells, the functions of the TME and the role of lactate interchange between cancer and non-cancer cells in vivo are still needed to deepen our understanding of the nature of cancer and to develop effective forms of therapy. The efficacy of newly designed specific anion transporters able to move lactate out of the extracellular space is currently under investigation. These novel compounds may be effective in Warburg or glutamine-dependent cells and in lactate-exploiting cells. 

## Figures and Tables

**Figure 1 cancers-12-03244-f001:**
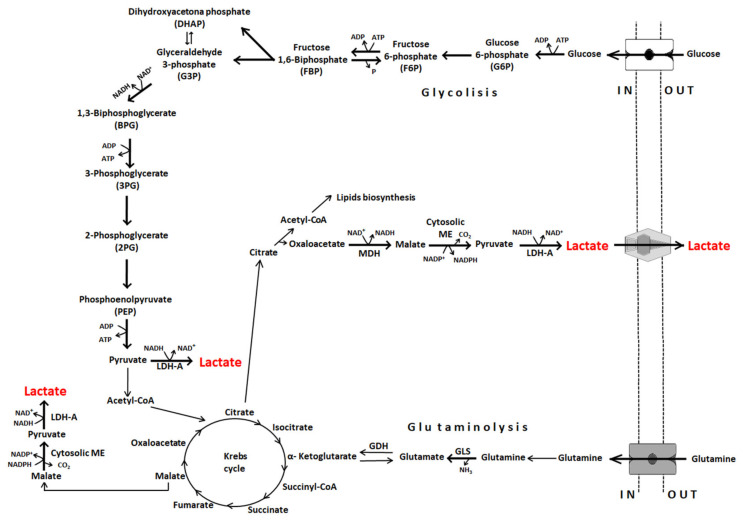
The main biochemical pathways implicated in cancer cell lactate generation. ME, malic enzyme; MDH, malate dehydrogenase; LDH-A, lactate dehydrogenase A; GLS, glutaminase; GDH, glutamate dehydrogenase.

**Figure 2 cancers-12-03244-f002:**
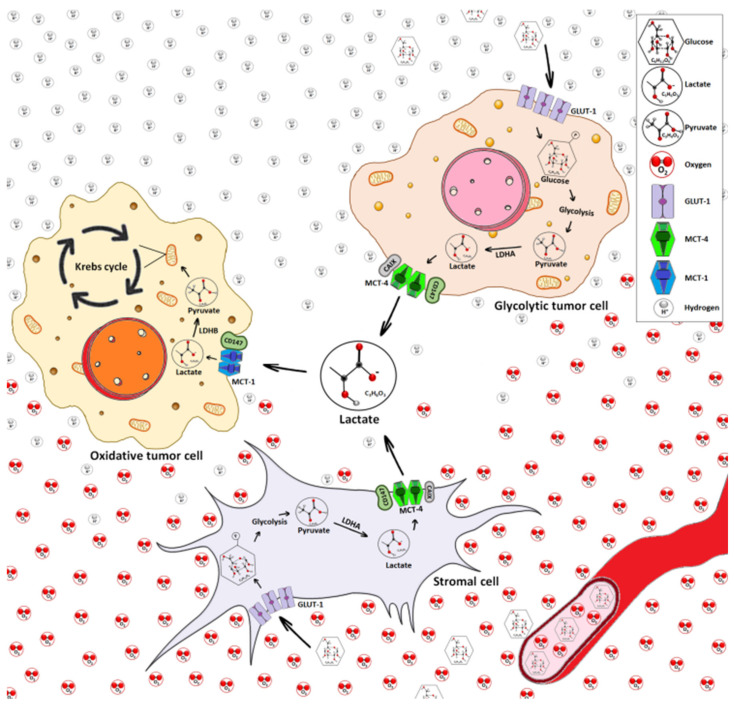
Intercellular lactate exchange in the tumor microenvironment (TME). The TME is a complex ultrastructure containing different cell types, including tumor cells, stromal cells, immune cells, blood vessels and cellular metabolites such as lactate. Oxidative tumor cells (orange nuclei) and stromal cells (gray cytoplasm) are supported by a favorable location with high nutritional and O_2_ availability. Oxidative tumor cells (orange nuclei) express MCT1 transporter which preferentially promotes lactate import. The glycolytic tumor cells (pink nuclei) produce lactate by the glycolytic pathway that culminates in the final reaction mediated by lactate dehydrogenase LDHA, and exhibit high expression of MCT4 favoring lactate export. Lactate can be used as an energetic source through its conversion to pyruvate via LDHB and then go to into the Krebs cycle for energy generation. Thus, the glycolytic tumor cell (pink nuclei), and stromal cell (gray) interchange of lactate with oxidative tumor cells (orange nuclei) increases tumor cell survival and proliferation. CD147: chaperone. CAIX: carbonic anhydrase IX. Made from the original idea [61].

**Figure 3 cancers-12-03244-f003:**
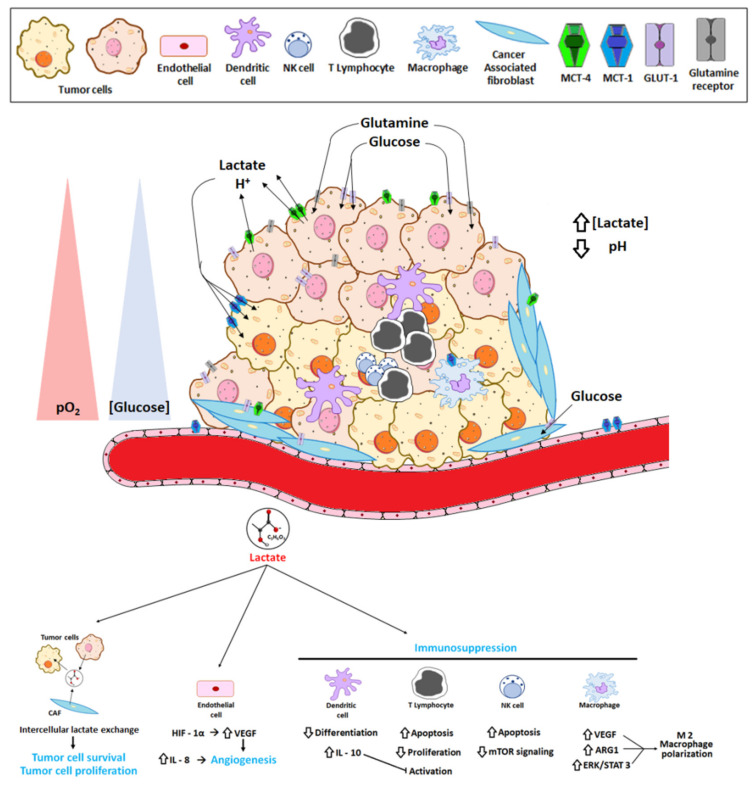
Effect of lactate on the tumor microenvironment (TME). Lactate secretion by tumors and stromal cells mainly acidifies the tumor microenvironment and creates a lactate interchange between them that increases tumor cell survival and proliferation. Lactate also stimulates tumor angiogenesis via endothelial cells and contributes to the immune scape by altering several immune infiltrating cells.

**Table 1 cancers-12-03244-t001:** Compounds to inhibit lactate production.

Target	Drug	Type of Cancer or Cell/Animal Model	Research Phase	References
LDH	FX-11	B-lymphoid cells (P493, P198) Xenograft model	Pre-clinical	[167]
	Gossypol AT-10	Multiple kinds of cancer	Phase I and Phase II clinical trials^a^	[171,172]
	Galloflavin	Liver cancer (PLC/PRF/5) Hepatocellular carcinoma	Pre-clinical	[168,174]
	N-hydroxyindole-based compounds	Colon (Caco-2, HCT116 and HT29) Bladder (5637, HT1197, HT1376, RT4, SW780, T24, TCCSUP and UM-UC-3)	Pre-clinical	[169]
	FX866	Pancreatic cancer (P198) Xenograft model	Pre-clinical	[167]
	Oxamate	Hepatocellular carcinoma Medulloblastoma	Pre-clinical	[174,175]
	2 Thio-6-oxo1,6-dihydropyrinidine (DHPMs)	Pancreatic carcinoma (MIA PaCa-2) Mouse model	Pre-clinical	[176]
	GNE-140	Colon adenocarcinoma (LS174T) Mouse model Pancreatic carcinoma (MIA PaCa-2)	Pre-clinical	[177,178]
	Pyzazole based inhibitors	Pancreatic carcinoma (MIA PaCa-2) A673 Sarcoma (A673)	Pre-clinical	[179]
	1-(Phenylseleno)-4-(Trifluoromethyl) Benzene (PSTMP)	Large cell lung cancer (NCI-H460) Breast cancer (MCF-7) Hepatocellular carcinoma (Hep3B) Malignan melanoma (A375) Colorectal adenocarcinoma (HT29) Murine lung cancer (LLC)	Pre-clinical	[180]
	Compound 11	Osteosarcoma (MG-63)	Pre-clinical	[181]
	Compound 24c	Pancreas carcinoma (MiaPaCa-2)	Pre-clinical	[182]
	Peptides collections (QLYNL, LIYNLL, IYNLLK, KWYNVA, and KVVYNV)	None	*In silico* modeling	[183]

Notes: a ClinicalTrials.gov identifier: NCT01791595.

**Table 2 cancers-12-03244-t002:** Compounds to inhibit lactate transport.

Target	Drug	Type of Cancer or Cell/Animal Model	Research Phase	References
MCT1	AR-C155858	Murine breast cancer (4T1)	Pre-clinical	[186]
	SR 13800	Burkitt lymphoma (Raji)	Pre-clinical	[185]
	AZD 3965	Human diffuse large B-cell lymphomas (HBL-1 and TMD8) Human B-cell lymphoma (WSU-CLCL-2 and SU DHL10) Lymphoblast (HT) B-cell non-Hodgkin lymphoma (Karpas-422 NHL) Raji Burkitt’s lymphoma cells	Pre-clinical Phase I/II of clinical trials ^a^.	[186,188,189]
	α- cyano-4-hydroxycinnamate (CHC)	Colorectal cancer (HCT15 and RKO) Murine cancer model	Pre-clinical	[82,192]
	BAY-8002	Hematopoietic malignancies, Raji, and Daudi Burkitt lymphoma cells	Pre-clinical	[193]
MTC4	Diclofenac	Caco-2 cell line	FDA- Approved as anti-inflammatory drug	[200]
	Bindarit	Xenopus oocyte	Experimental Research	[201]
	AZ93	Wide range of cancer cells	Pre-clinical	[202]
MCT1/MCT4	Syrosingopine	HeLa, HAP1, HL60 cells, liver tumor mouse model	Pre-clinical	[189]
	Lonidamine	DB-1 melanoma cell	Pre-clinical Phase III of clinical trials (prostate cancer) ^b^	[203]
CD147	pCMBS	Molecular biology (Xenopus oocyte, murine cells)	Experimental Research	[205]
	AC-73	Hepatocellular carcinoma (SMMC-7721, Huh7) Orthotopic transplant nude mouse model	Pre-clinical	[206]
Metuzumab	Xenograft models (A549, NCI-H520) Monkey model	Pre-clinical	[207]

Notes: ^a^ adult glioblastoma, phase II, ClinicalTrials.gov identifier: NCT00540722; lymphoma, phase II, ClinicalTrials.gov identifier: NCT00275431; adrenocortical carcinoma, phase II, ClinicalTrials.gov identifier: NCT00848016; leukemia, phase II, ClinicalTrials.gov identifier: NCT00286780; laryngeal cancer, phase II, ClinicalTrials.gov identifier: NCT01633541; small-cell lung cancer, phase II, ClinicalTrials.gov identifier: NCT00773955; prostate cancer, phase II, ClinicalTrials.gov identifier: NCT00666666; ^b^
ClinicalTrials.gov identifier: NCT00435448.

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
