# Peer review of "Lactate in the Tumor Microenvironment: An Essential Molecule in Cancer Progression and Treatment"

_cancers, 2020, doi:10.3390/cancers12113244_

Round 1

Reviewer 1 Report

This is a nicely written and comprehensive review on the role of lactate in the TME and its versatile functions on tumor and immune cells.

My only recommendation for improvement of this paper is to enlarge the figures and tables in order to make them easier to read.

Author Response

Thank you very much for your comments and suggestions.

As the reviewer suggests we have enlarge figures, tables and legends in order to improve the manuscript and for clarity for the future readers.

Reviewer 2 Report

I enjoyed reading this comprehensive review and feel that it is a timely and useful addition to the field.

The section on potential therapies would benefit by 1-2 subtitles, grouping the type of therapies together a bit better. 

I feel that the authors could comment on how much of the data is derived from subcutaneous mouse tumors which are likely providing a skewed set of data compared to human tumor data.  Are there differences they can comment upon?

Minor:  The text (and overall plan) of Figs. 1 and 2 and bottom part of Fig. 3 seem much  too small to read easily.

Author Response

Thank you very much for your comments and suggestions; they help us to improve the manuscript.

In reference to adding 1-2 subtitles to the section of potential therapies as the reviewer suggests, we believe that the current 2 sections (Targeting lactate production and Targeting lactate transporters) is enough to  give at the future readers a wide vision of the current therapies. In our opinion adding subtitles might not improve too much the information that we want to transmit. Of course, there are others ways to organize the manuscript section.

In reference to the reviewer comment “how much of the data is derived from…” : In table I and II we provided information about type of cancer or cell/animal models, as well as, research phase of each drug. In our opinion the information provided already clarifies the information requested by the reviewer.

As the reviewer suggests we have enlarge figures, tables and legends in order to improve the manuscript and for clarity for the future readers.

Reviewer 3 Report

The manuscript by Pérez-Tomás and Pérez-Guillén discusses the role of lactate in tumor progression with a focus on its effect on tumor microenvironment (TME). Lactate is being increasingly recognized as an important oncometabolite which aids tumor development by promoting tumor growth and metastasis, evasion of immune response and therapy resistance. The manuscript provides an introduction to lactate metabolism and transport followed by a review of regulation of lactate synthesis by oncogenes and tumor suppressors and inverse regulation of oncogenic drivers by lactate. Furthermore, the manuscript discusses how high lactate concentrations in the TME drive ‘immune escape’ by suppressing immune cells while promoting development of immunosuppressive cells in the TME. The manuscript also reviews literature covering how lactate promotes metastasis and resistance to therapy. Finally, several strategies currently in consideration which target lactate production, transport and exchange for cancer therapy have been discussed. The manuscript is well written and structured and provides a comprehensive and thorough understanding of the role of lactate in cancer.

Minor suggestions and comments include:

  • Table 1 is missing and Table 2 is shown in place of Table 1.

  • Figure 2 legend line 173 mentions that “Glycolytic tumor cells (pink nuclei) and stromal cells (gray cytoplasm) are supported by a favorable location with high nutritional and O2 availability.” This is incorrect as glycolytic cells are typically hypoxic while oxidative cells are in proximity to blood vessels and high O2 availability.

  • Considering the title “Lactate in the tumor microenvironment...” I would expect a more detailed discussion on lactate exchange between the tumor cells and the stroma particularly CAFs.

  • The figures in the manuscript can be made larger.

  • The font sizes in the figures and tables can be made larger for clarity.

  • The word ‘harmful’ is written twice in Line 319.

  • There should be a period (.) after ‘understood in line 401.

  • The words ‘and also’ should be removed from line 547.

Author Response

Thank you very much for your comments and suggestions; they help us to improve the manuscript.

Point 1- As we are checked, in the manuscript both tables are in their correct positions.

Point 2- As the reviewer suggested, we have corrected figure 2 legends where a mistake was made so the sentence is now correct.

Point 3- In reference of the reviewer suggestion of including “more detailed discussion on lactate…”: We agree that this point is important, in fact, the recent review of Luigi Ippolito et al. (Trends in Biochemistry 2019), discuss this point in deep as well as other reviews. This, together with the limited space suggested by the journal contributed to not going that deep into this particular point and cited the main reference and concepts regarding lactate exchange.

Point 4 and 5- As the reviewer suggests we have enlarge figures, tables and legends in order to improve the manuscript and for clarity for the future readers.

Point 6- The extra word harmful has been remove from the text.

Point 7- We added a period (.) as the reviewer suggested.

Point 8- Both words (and also) have been removed from remove from the text as the reviewer suggested.

Reviewer 4 Report

The manuscript "LACTATE IN THE TUMOR MICROENVIRONMENT: AN ESSENTIAL DRIVER IN CANCER PROGRESSION" is another review on the role of lactate in tumor progression. While the topic is very interesting it is not clear how much new information this review is adding to the existing literature of several recent reviews on the same topic. 

I checked similar publication and from the first 10 publications only two were cited. Most of the newest literature is missing: e.g. doi: 10.3389/fonc.2019.01143; 10.3389/fonc.2020.00231, 10.1111/pcmr.12495, 10.1016/j.semnephrol.2019.04.007; 10.3390/cancers11060750, 10.1016/j.gendis.2017.02.003.
Some citations are wrong  or misleading: Sentence in line 417/18 was nearly completely copy and pasted from Payen et al. [143] but the citation is for [147].

Author Response

Thank you very much for your comments and suggestions.

With this manuscript we are trying to provide a comprehensive and thorough understanding of the role of lactate in cancer and we realize that other reviews related with lactate have been published. Nevertheless, we want to highlight from our manuscript an interesting introduction of lactate metabolism, transport and synthesis regulation. Actually, we added some original and useful images in order to provide an easiest understanding of what we want to transmit to the future readers. Furthermore, we describe in deep how high lactate concentrations in the TME drive “immune escape” by suppressing immune cells while promoting development of immunosuppressive cells.

The manuscript also reviews ¡s literature covering how lactate promotes metastasis and resistance to therapy. Finally, therapeutic strategies have been considered. In table I and II we provided information about type of cancer or cell/animal models, as well as, research phase of each drug.

Of course, is important taking into account that there is a limited space suggested by the journal and based on that we couldn’t put add as many references as we wanted to. However in our reference section we have: 29 references from 2019, 15 references from 2018, 29 references from 2019 and finally 7 references from 2020. We have revised carefully the citations through the text in order to avoid misleading.

In reference to the sentence line 417/18 that the reviewer suggested that was “nearly completely coy and pasted from Payen et al[143] but the citation is for [147]”. We revised carefully this sentence and we found out in reference 147 section 4.2 (Kumar, D et al. Life Sciences. 2019.) of the manuscript the following sentence: “the role of CD147 as the counter receptor, thereby, facilitating the extracellular matrix degradation, tumor cell invasion and metastasis [46]. CD147 creates a positive feedback loop for its own expression, which ends up in regulation of MMPs, cytokines and CD147 itself [7].” Based on this sentence and other through the text and their reference 46 (J. Sun, M.E. Hemler, Regulation of MMP-1 and MMP-2 production through CD147/ extracellular matrix metalloproteinase inducer interactions, Cancer Res. 61 (5) (2001) 2276–2281.) we reach the idea that we express in the manuscript with some similarity showed in the Payen manuscript (143) but in any case seems a copy paste.

Round 2

Reviewer 4 Report

The manuscript improved satisfactory and can be accepted. There is some small typos to be corrected.

Fig. 1 it must be glycolysis

L.116 must be "due to poorly perfused 116 dense tumor tissue,"